# Comparison of TENS electrodes and textile electrodes for electrocutaneous warning

**Eva-Maria Dölker**[1]*, **Yasemin Cabuk**[1], **Tino Kühn**[2], **Jens Haueisen**[1]

**1** Institute for Biomedical Engineering and Informatics, Technische Universität Ilmenau, Ilmenau, Germany, **2** Institute of Manufactoring, Technische Universität Dresden, Dresden, Germany

\* eva-maria.doelker@tu-ilmenau.de

**Data availability statement:** All data are in the paper and/or supporting information files at https://doi.org/10.5281/zenodo.14833284.

## Abstract

Electrocutaneous stimulation can be employed to alert workers in potentially hazardous situations. Previous parameter studies used TENS electrodes during the developmental process of the electrical warning system. As a step towards more practicability, we now focus on electrocutaneous stimulation through wearable textile electrodes. In order to determine the feasibility of a novel textile electrode cuff in comparison to previously used TENS electrodes, two studies were conducted. In a study on $n = 30$ participants, perception, attention, muscle twitch, and intolerance thresholds as well as qualitative and spatial perceptions, were determined for eight pairs of electrodes circumferentially placed around the upper right arm for TENS and for textile electrodes. In a second study on $n = 36$ participants, these thresholds were also determined during vibration, and a warning signal pattern was presented during vibration. We found smaller perception thresholds for the textile electrodes in comparison to the TENS electrodes for all 8 electrode pairs and occasional differences for the attention and intolerance thresholds, which might be mainly explained by the varying electrode sizes due to the manual production process of the textile electrodes. Stimulation using textile electrodes within the cuff showed less frequent muscle twitches compared to TENS electrodes. Other qualitative and spatial perceptions appeared comparable. The perception, attention, and intolerance thresholds increased during vibration comparable to previous results with TENS electrodes. The feasibility of using the textile electrodes during vibration and for the application of a warning signal was successfully demonstrated. Occasional cases occurred where the transition impedance was too high while using the textile electrodes. Future studies will focus on electrode optimization to achieve a wearable solution with both low electrode-skin transition impedance and minimal muscle twitching.

## Introduction

In occupational safety, reliably warning workers in hazardous situations is crucial. Warning signals should convey different meanings and urgency levels. For example, a railway track worker must be reliably informed about an approaching train [1], including which track it's on and its distance. Current methods use acoustic [1,2] or visual signals [3], which can fail under loud or low-visibility conditions. Therefore, we aim to develop a warning system using

**Funding:** We would like to thank the Free State of Thuringia for its support under project number 2018 IZN 004 co-financed by the European Union under the European Regional Development Fund (ERDF). This work was supported by a scholarship for the professional qualification of young female scientists, funded by the Free State of Thuringia. We acknowledge support for the publication costs by the Open Access Publication Fund of the Technische Universität Ilmenau.

**Competing interests:** The authors have declared that no competing interests exist.

electrocutaneous stimulation through wearable textile electrodes. This system should convey information through varying the amplitude, frequency, or spatio-temporal coding of the warning signal.

Considering this long-term goal, we envisioned the following general requirements: The signal should be perceivable, distinct, and unique, capable of conveying a few different types. Electrodes should be comfortably placed with minimal sweat, motion restriction, and risk of muscle contraction, and should not interfere with other safety equipment. The system should comply with ISO EN 60601 standards [4] and be user-friendly: well-tolerated, wearable for up to 8 hours, lightweight, integrated into work clothes, and have stable electronics and power.

Noninvasive electrical stimulation is employed for a variety of purposes, ranging from general applications such as muscle stimulation [5–11] and transcutaneous electrical nerve stimulation (TENS) [12–14] to more specialized uses [15–19]. Specialized uses of electrocutaneous stimulation include transcutaneous electrical acupoint stimulation [15,17, 18], neuromodulation for treatment of patients with amyotrophic lateral sclerosis [20], the investigation of experimental knee pain on locomotor biomechanics [19], or the study of hyperalgesia induced by electrical stimulation [16]. Research on electrical muscle stimulation has revealed its significance in several areas, including the relationship between muscle length and torque generation [6], its influence on subjects with cerebral palsy, multiple sclerosis, and stroke on upper motor neuron syndrome symptoms [9], its influence on visual sensory reweighting [5], and its role in assessing muscle metabolism [7]. Additionally, a systematic review by Ibitoye highlights the effectiveness of leg exercises supported by muscle stimulation promoting muscle and bone health in paralyzed lower limbs [8]. TENS, on the other hand, has been extensively studied for its effectiveness in pain relief [12,13] and its potential to reduce postoperative wound infections [14].

Electrocutaneous stimulation is also applied in medical prosthetics, where it is used to deliver sensory feedback [20–22]. However, the design and functionality of electrocutaneous stimulation differ significantly in warning systems, as these require higher amplitude signals [23] compared to the lower intensities used in prosthetic sensory feedback. As a result, parameter studies are essential for the development of an electrical warning system.

In a previous study with 81 participants [23], we defined three key thresholds for electrical warnings: the perception threshold ($A_p$) for just noticeable stimuli, the attention threshold ($A_a$) for attention-drawing stimuli, and the intolerance threshold ($A_i$) for intolerable stimuli. We investigated these thresholds along with qualitative and spatial perceptions, varying pulse widths, electrode sizes, and positions. Threshold values decreased with increasing pulse width and increased with larger electrode sizes (15 mm × 15 mm to 40 mm × 40 mm). Knocking was the main perception at perception and attention thresholds, while muscle twitching, pinching, and stinging were reported at the intolerance threshold. Stimulation was perceived within the region of the used electrode pair. Further studies on 94 participants assessed the impact of vibration, showing increased perception thresholds with increasing vibration amplitude and frequency [24]. Another study with 52 participants in different climate conditions found that perception thresholds increase with decreasing temperature, and muscle twitch thresholds rise with increasing temperature, with no significant humidity effect [24]. Women showed smaller perception thresholds and less muscle twitching than men.

We used TENS electrodes in the above cited previous studies because these electrodes typically provide low electrode-skin transition impedances. In a first comparison of TENS electrodes and textile electrodes study [25] with $n$ = 30 participants, we compared perception, attention, and intolerance thresholds as well as the qualitative and spatial perceptions for

three electrode pairs placed at the lateral side of the upper right arm. We found no significant differences, but the textile electrodes showed elevated electrode-skin transition impedance.

The aim of the current study is to compare a novel set of 8 textile electrode pairs mounted within a novel cuff with the previously used TENS electrodes. For that purpose, two studies were conducted. In the first study ($n = 30$), the perception thresholds, qualitative and spatial perceptions were compared between textile and TENS electrodes. In the second study ($n = 36$), the textile electrodes were investigated during vibration and under the presentation of a warning signal to investigate whether the results were comparable to the ones obtained with the TENS electrodes.

The paper is organized as follows. In section Materials and methods, we present the study groups of the two studies, the experimental setup for electrocutaneous stimulation, properties of TENS and textile electrodes, definition and determination of thresholds, qualitative and spatial perceptions, and the application of vibration. In this section, also the experimental paradigms of both studies are described. The section is finished with a description of the statistics. The section Results shows perception, attention, muscle twitch, and intolerance thresholds as well as qualitative and spatial perceptions for TENS and textile electrodes of study 1 and the same thresholds for the textile electrodes during rest and under vibration of study 2. Additionally, the results of the application of a warning signal during mechanical vibration are presented. Results are discussed in the Discussion section. In the last section, we give the conclusions and directions for future work.

## Materials and methods

### Study group

The descriptive statistics of the two study groups are listed in Table 1, where $n = 24$ participants attended both studies. For $n = 15$ participants, study 1 and study 2 (Table 1) took place at the same day. Participants have been recruited between 05.05.2023-12.07.2023. The ethics committee of the Faculty of Medicine of the Friedrich-Schiller-University Jena, Germany, approved the studies. All methods were carried out in accordance with relevant guidelines and regulations. All participants gave written informed consent.

For the day of the experiment and the day before the experiment, participants were asked to get a sufficient amount of sleep, to not consume caffeine, nicotine, or alcohol, to drink enough (approx. 2 l), to do no hard, physical work or sports, and not treat upper arms with skin cream.

### Experimental setup

**Electrical stimulation.** The experimental setup was identical to the one in our previous publication [23]. Here, we give a brief overview, for details please see [23]. We use an in-house implemented program written in LabVIEW 2017 (National Instruments, Austin, TX, USA)

**Table 1. Descriptive statistics of the study groups.**

| Property | Study 1: Comparison of TENS electrodes vs. textile electrodes | Study 2: Use of textile electrodes under vibration |
|---|---|---|
| Participants | $n = 30$ | $n = 36$ |
| Gender | female: 12, male: 18 | female: 11, male: 25 |
| Age | 25 years $\pm$ 3 years (mean $\pm$ standard deviation) | 25 years $\pm$ 3 years (mean $\pm$ standard deviation) |
| Handedness | right-handed: 28, left-handed: 2 | right-handed: 35, left-handed: 1 |
| Arm circumference (right arm) | $30.9 \pm 4.8$ cm (mean $\pm$ standard deviation) | $31.3 \pm 5.2$ cm (mean $\pm$ standard deviation) |

to control a constant current stimulator DS5 (Digitimer Ltd, Letchworth Garden City, UK) via a PC and a multiplexer D188 (Digitimer Ltd, Letchworth Garden City, UK), which activates one of its 8 output channels and thereby delivers the stimulus to one of the desired electrode pairs. The biphasic rectangular current stimulation signal is defined by the parameters: amplitude $A$, pulse width $t_p$, pulse interval (1/pulse frequency $f_p$) and number of pulses $n_p$. The DS5 measures the impedance in order to calculate the voltage that is necessary to provide the desired current, which is up to 25 mA in the experimental setup for electrical warning. If the impedance (e.g. due to high transition impedances between electrode and skin) is too high, a voltage >100 V would be necessary to provide the desired current of 25 mA (or smaller). In that case, the DS5 gives an alarm beep tone and the applied stimulation current cannot be increased further for safety reasons.

**TENS electrodes.** The 16 electrodes (re-useable self-adhesive TENS electrodes, axion GmbH, Leonberg, Germany, size 25 mm × 40 mm) were placed pair-wise along the centerline between the shoulder joint and the elbow of the right arm. In the circumferential direction, the electrodes were placed at distances of 1/8 of the arm circumference. For each electrode pair, one electrode was placed 5 mm above the center-line and the other 5 mm below. The electrode pairs were numbered consecutively, where electrode pair 1 corresponded to the anterior, 3 to the lateral, 5 to the posterior, and 7 to the medial position of the arm. Electrode pairs were arranged in a vertical configuration with a top and a bottom electrode. The TENS electrodes are larger than the textile electrodes as a smaller size of the TENS electrodes led to a too high electrode-skin transition impedance where voltages > 100 V would have been necessary to stimulate with amplitudes up to 25 mA.

**Textile electrodes.** The electrodes of the textile cuff consist of silver-coated knitted fabric covered by carbon black filled silicon. The textile cuff is stretchable and thus the horizontal distance between the electrodes changes in dependence of the arm circumference. However, as the stretchability is limited, four different cuffs with textile electrodes were manually manufactured (BORN GmbH, Dingelstädt, Germany) covering the clothes sizes S - XL. Fig 1 shows the cuff of size S. The cuff is selected according to the participant's arm circumference (cf. Table 2). The minimal vertical distance between two electrodes, the minimal horizontal distance between two electrodes, and the electrode dimensions are given in Table 2. The size and distance variability is caused by the manual production process.

**Thresholds, qualitative and spatial perceptions.** To find an operating range for an electrocutaneous warning system, three thresholds were estimated [23]: (1) a just noticeable stimulus defines the perception threshold $A_p$, (2) a stimulus drawing attention to itself defines the attention threshold $A_a$, and (3) a stimulus generating intolerable perceptions defines the intolerance threshold $A_i$. The attention threshold acted as an indirect intensity measure for the perception of an attention-attracting stimulation. In addition to these three thresholds, the muscle twitch threshold $A_m$ was determined which marks the minimal amplitude where the participant perceives muscle twitches.

To determine these thresholds, a single biphasic stimulation pulse of pulse width $t_p = 150\,\mu s$ was applied. The amplitude was increased gradually from 0 mA up to a maximum of 25 mA by steps of 0.1 mA, 0.2 mA, or 0.5 mA. The steps were chosen adaptively by the trained operator. Before the start of the experiment, the meaning of the three thresholds was explained to the participant. The operator always told the participant which threshold would be determined next such that the participant could focus on the threshold determination.

At each threshold, except for muscle twitch, the operator asked the participant about qualitative and spatial perception. At the muscle twitch threshold the participant was only asked for the spatial perception. For the qualitative perception, there were the following choices in the questionnaire: knocking, scratching, stinging, pain, muscle twitch, tickling, itching,

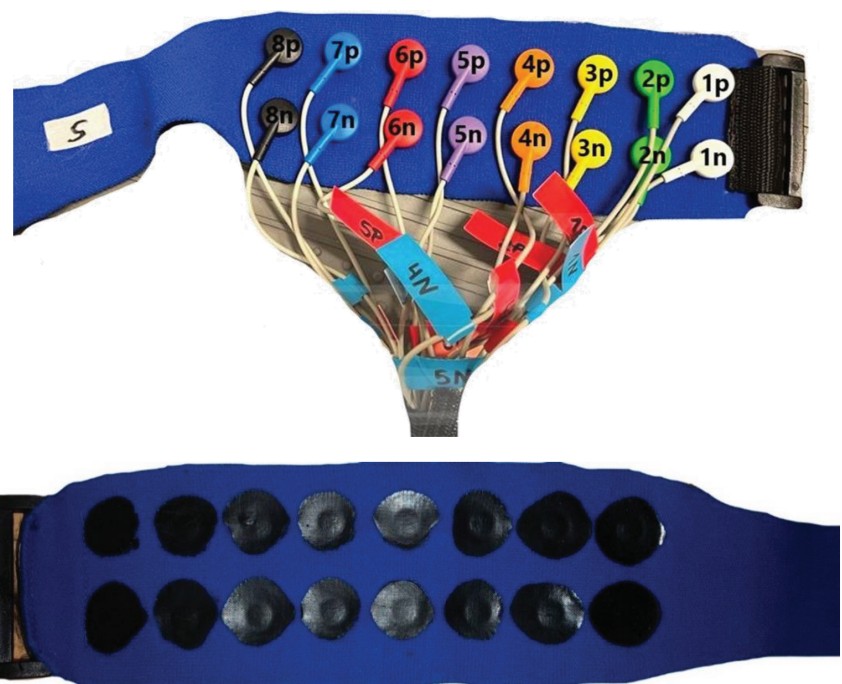

**Fig 1. Electrode configuration of the textile cuff of size S.** Top: External view with numbered electrode pairs, Bottom: Internal view of the electrode pairs consisting of silver-coated knitted fabric with carbon black filled silicon coating.

**Table 2. Size chart of the textile electrode cuffs.**

| Cuff size | S | M | L | XL |
|---|---|---|---|---|
| Arm circumference (cm) | ≤ 28 | >28–33 | >33–40 | >40–46 |
| Minimal horizontal distance (mm) | 5.07 ± 1.67 | 2.32 ± 1.48 | 9.00 ± 1.73 | 16.36 ± 2.02 |
| Minimal vertical distance (mm) | 10.00 ± 0.71 | 7.75 ± 1.48 | 9.50 ± 1.00 | 8.88 ± 0.93 |
| Electrode size (mm×mm) | 19.88 ± 2.45 × 19.13 ± 0.78 | 21.56 ± 4.12 × 21.75 ± 1.75 | 20.44 ± 1.12 × 19.44 ± 0.70 | 21.13 ± 1.45 × 19.63 ± 0.99 |

The cuff was designed according to the individual pressure and comfort requirements, related to the upper arm shape. Distances and size are given as mean± standard deviation to created four uniform standards. The minimal distances describe the edge-to-edge distances of the electrode coatings for a non-stretched cuff: horizontal for neighboring electrodes, vertical for one electrode pair (cf. Fig 1). The electrode size parameter describes the electrode coating in vertical and horizontal directions which is directly contacting the skin. (cf. Fig 1)

pinching, squeezing. It should be noted that a painful perception is not necessarily identical to the intolerance threshold. Reaching the intolerance threshold means that the participant perceives the stimulus as intolerable, whereas the assigned qualitative perception is selected from the questionnaire and could be e.g. stinging, pinching, or also pain. A painful perception could also happen before reaching the intolerance threshold when the perception is painful but not intolerable. The categories of qualitative perception were chosen according to our previous research [23,26] and based on other fields of electrocutaneous stimulation [27–31]. The choices for the spatial perception included: between the electrodes, at both electrodes, at the upper or lower electrode, extending beyond electrodes, at other parts of the body.

The above-mentioned definitions of the thresholds and the perception categories of the questionnaire were explained to the participant and queries from the participant were answered before starting the experiment.

**Vibration.** The vibration was transferred via the hand onto the arm using a vibroshaper (MediaShop GmbH, Neunkirchen, Germany). The use of the vibroshaper enables the adjustment of the vibration frequency in steps between 1 and 99 (equivalent from 8 to 13 Hz). The position of the hand influences the transferred vibration amplitude from 0 to 8 mm. The influence of the vibration on the electrocutaneous perception was investigated for the frequency of 9.5 Hz and the vibration amplitude of 8 mm (cf. Fig 2) as the highest vibration frequency and amplitude that was used in our previous studies [24].

### Experimental paradigm

**Preparation.** The devices were switched on 30 min before the actual experiment for warming up. Additionally, the current stimulator DS5 was stabilized. For that purpose, a load resistor (1 kΩ) was connected to the DS5, and a series of stimulation pulses were sent through the load resistor multiple times. To lower the electrode-skin transition impedance between the skin and the electrodes, the right upper arm of the participant was cleaned and moistened with a wet towel. If the textile electrodes were used, the upper right arm was sprayed with warm 0.9 % NaCl solution. The participant was asked to sit in a comfortable position, relax the arm, keep the view away from the experimental devices, and concentrate on the perception of the stimulation. During the experiment, the arm was positioned in a relaxed bent position on a table except if the Vibroshaper (cf. Fig 2) was used. The experiment was conducted at a room temperature of approx. 23°C.

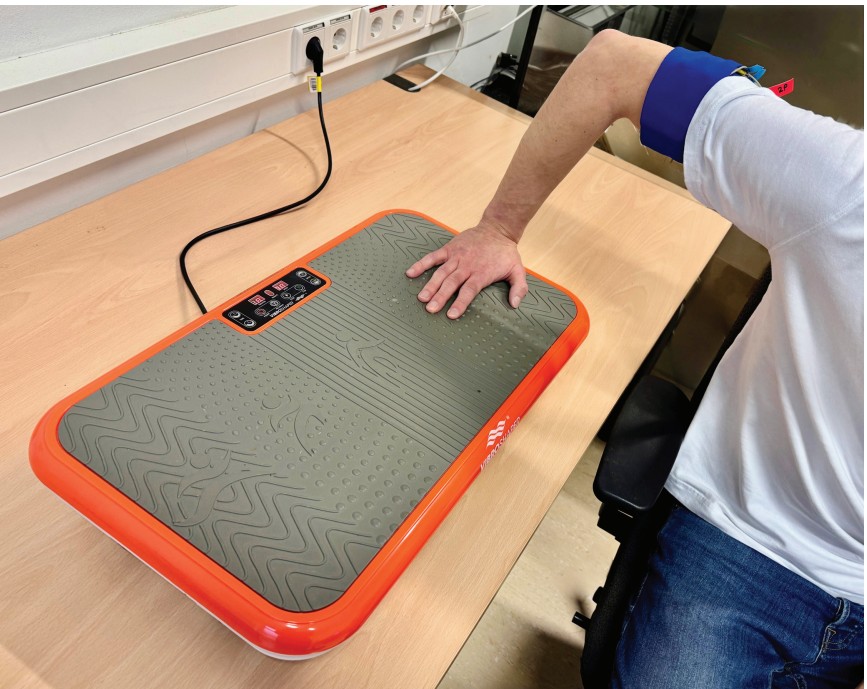

**Fig 2. Hand position on the Vibroshaper during electrocutaneous stimulation at the upper right arm.** A vibration amplitude of 8 mm and a frequency of 9.5 Hz were used.

**Study 1: Comparison of TENS electrodes vs. textile electrodes.** *Reference threshold experiment with TENS electrodes.* During the reference threshold experiment, the threshold determination procedure was repeated ten times for electrode pair 3. Using the repetitive threshold determination, the participant was able to learn the detection of the thresholds and the reporting of the perceptions. It ensured that the participant got used to the novel feeling of current stimulation. The mean values for perception, attention, muscle twitch and intolerance thresholds were calculated from the last 3 measurement series. Regarding the qualitative and spatial perception, the most frequent out of the 3 measurement series were used. If three varying perceptions occurred the last one was selected.

*Thresholds, qualitative and spatial perception for TENS and textile electrodes.* The perception, attention, muscle twitch, and intolerance thresholds as well as the qualitative and spatial perceptions were determined for the TENS electrode pairs 1 to 2 and 4 to 8. After that the textile cuff was selected according to the arm circumference (cf. Table 2) and attached. Thresholds, qualitative and spatial perceptions were determined for textile electrode pairs 1 to 8. If Out-Of-Compliance Errors occurred, the threshold determination at this electrode pair was stopped and the corresponding amplitude was noted.

**Study 2: Use of textile electrodes under vibration.** *Reference threshold experiment with cuff electrodes.* The cuff was selected according to the arm circumference (cf. Table 2). The reference threshold determination at electrode pair 3 was conducted analogously to the one in study 1. An individual warning amplitude $A$ was calculated as

$$A = A_a + 0.2 \cdot (A_i - A_a), \tag{1}$$

where $A_a$ and $A_i$ denote the mean values of the attention and intolerance thresholds at electrode pair 3.

*Thresholds for textile electrode pair 3 under vibration.* The thresholds $A_p$, $A_a$, $A_m$ and $A_i$ were determined under vibration with an amplitude of 8 mm and a frequency of 9.5 Hz for electrode pair 3. A second warning amplitude $A_v$ was determined according to (1).

*Warning pattern.* In a previous pilot study (n = 16) [32], circumferential electrical stimulation signals around the arm (pair 1 to 7) creating a vibrating sensation were designed as warning patterns. The pattern with the highest alertness levels is used now in this study. This study aimed to investigate the co-perception of this warning signal during mechanical vibration under the use of novel textile cuff electrodes. First, five consecutive bi-phasic rectangular stimulation pulses (amplitude $A$, pulse width $t_p$ = 150 μs) were sent with an initial pulse interval of 19 ms [32] in order to achieve a vibrating sensation. The pulse interval was adjusted if the participant perceived a pulsating or continuous sensation until a vibrating sensation was reported. The circumferential warning signal was delivered sequentially from electrode pairs 1 to 7, with each pair stimulated for 0.4 seconds and a 0.1-second break between pairs. The number of pulses per pair was based on the individual pulse interval. Participants rated the stimulation signal on alertness, discomfort, and urgency scales from 0 (no perception) to 9 (very strong). Initially, the warning pattern was tested under vibration with warning amplitude $A$ receiving a score of 5 on all scales. The participants then rated subsequent signals relative to this baseline. The stimulation amplitude was increased by steps of 2 mA until $A_v$. If it was still acceptable for the participant, the amplitude was further increased by steps of 1 mA up to a maximum of 25 mA. The presentation of the warning pattern was stopped at any amplitude where the stimulation was too uncomfortable for the participant. If too high electrode-skin transition impedance occurred during the warning pattern presentation, the corresponding electrode pair was rejected from the circumferential pattern 1-2-3-4-5-6-7. If more than two electrode pairs needed to be rejected, the pattern presentation was stopped, as

minimal 5 consecutive electrode pairs are necessary for a circumferential warning pattern to achieve a stable alertness [32].

## Statistics

The experimental results were analyzed using MATLAB 2023b® (The MathWorks, Inc., Natick, Massachusetts, USA). The results are visualized using boxplots and quantified by median M and the interquartile range *IQR* due to skewed distributions with outliers. The thresholds in dependence of electrode type are compared by paired Wilcoxon tests. *p*-values are corrected separately by the Bonferroni-Holm procedure for each investigated threshold. *p*-values < 0.05 are considered statistically significant.

## Results

### Study 1: Comparison of TENS electrodes vs. textile electrodes

Fig 3 shows the perception, attention, muscle twitch, and intolerance thresholds as boxplots in dependence of the electrode pair and the electrode type comparing TENS and textile electrodes. Median perception thresholds appeared smaller for textile electrodes compared to TENS electrodes (*p*<0.05) for all electrode pairs. Median attention thresholds showed statistically significant differences for electrode pairs 5 (*p* = 0.036) and 7 (*p* = 0.019). No significant differences were found regarding the muscle twitch thresholds. Median intolerance thresholds showed statistically significant differences for electrode pairs 3 (*p* = 0.048) and 4 (*p* = 0.023). The number of participants *n* was smaller than 30 for attention, muscle twitch, and intolerance thresholds as indicated in Fig 3b–3d, with S1 Table–S3 Table showing the corresponding quantification. S1 Table shows the number of participants without muscle twitches that was larger for textile electrodes compared to TENS electrodes, except for electrode pair 4. S2 Table shows the number of participants that did not reach the intolerance thresholds until 25 mA in dependence of the electrode type and pair. The amount was slightly higher for TENS electrodes compared to textile electrodes. The number of participants where the attention and intolerance threshold could not be determined due to too high electrode-skin transition impedances using the textile electrodes is shown in S3 Table. Overall, too high transition impedance occurred for 9 participants: 3 participants with cuff size S, 4 with M, 1 with L, and 1 with XL. A histogram of used cuff sizes within the study group is shown in S1 Fig. In two cases, cuff size S showed too high impedance at 4 electrode pairs. In all other cases, amount of electrode pairs with too high impedances were two or less.

Fig 4 shows the stacked frequency distributions of qualitative perceptions for perception, attention, and intolerance thresholds in dependence of the electrode pair and type. Overall, the most common qualitative sensation at the perception threshold was 'Knocking'. For electrode pairs 1 to 6, 'Knocking' was the dominant qualitative sensation at the attention threshold for both electrode types. At the medial positioned electrode pairs 7 and 8, it was 'Muscle Twitch' for TENS electrodes and 'Knocking' for textile electrodes. The most frequently reported qualitative sensations at the intolerance threshold were 'Stinging' and 'Muscle Twitch'. Muscle twitch occurred more frequently for the TENS electrodes compared to the textile electrodes and most often at medial electrode pairs 7 and 8.

Fig 5 shows the stacked frequency distributions of spatial perceptions for perception, attention, and intolerance thresholds in dependence of the electrode pairs and type. The electrical stimulation was perceived at the stimulated electrode pair. The option 'At another part of the body' was only chosen by the participants if muscle twitches occurred. Muscle twitches at 'Another part of the body' were reported at the upper arm, forearm, biceps, triceps, the

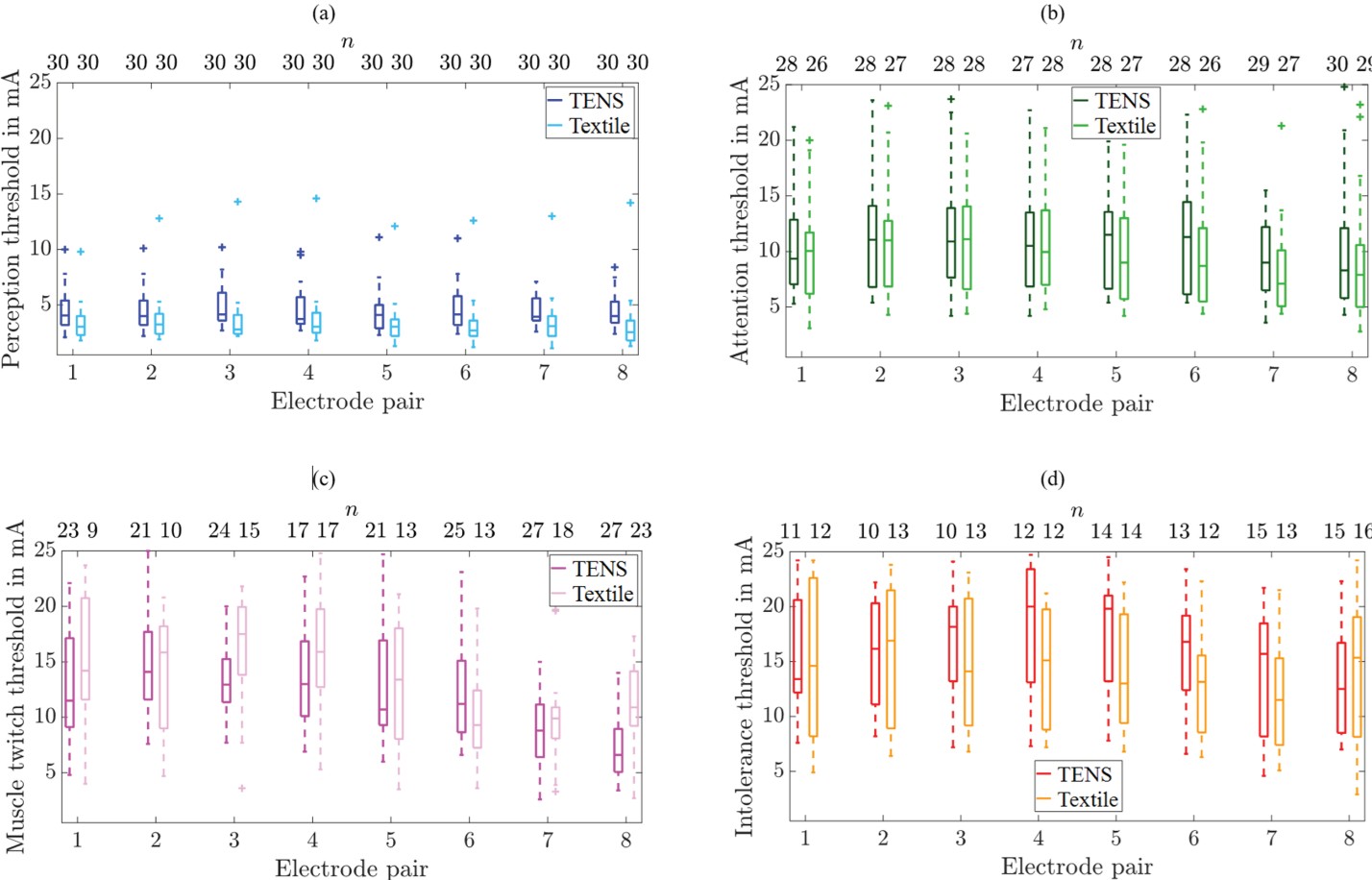

**Fig 3. Boxplots of perception (a), attention (b), muscle twitch (c), and intolerance thresholds (d) in dependence of electrode pair and type (TENS vs. textile).** Number of participants ($n$) out of 30 is given above each threshold. Each box plot shows summary statistics of threshold values for the study group. The solid horizontal line in each box represents the median, which, if off-center, indicates a skewed distribution. The box's bottom and top mark the 25th and 75th percentiles, respectively, with their distance defining the interquartile range. Dashed-line whiskers extend from the box to the furthest non-outlier value. Outliers are data points more than 1.5 times the interquartile range from the box's edges [33–35].

shoulder, the elbow, arm & hand, and arm & hand & finger. Differences between TENS and textile electrodes are related to the occurrence of muscle twitches (cf. Fig 4).

## Study 2: Use of textile electrodes under vibration

Fig 6 shows the perception, attention, muscle twitch, and intolerance thresholds as boxplots at electrode pair 3 for textile electrodes during rest and under vibration (amplitude: 8 mm, frequency: 9.5 Hz). The median values of perception, attention, and intolerance thresholds were higher during vibration than during rest ($p < 0.05$). No significant changes were found for the muscle twitch threshold. For one participant, the experiment was stopped after the first threshold determination during rest at electrode pair 3, as he was not feeling well on that day. There were no excessive electrode-skin transition impedances during the threshold determinations during rest and under vibration. The number of participants $n = 36$ was reduced for attention, muscle twitch, and intolerance thresholds (Fig 6) as thresholds were >25 mA or no muscle twitching occurred.

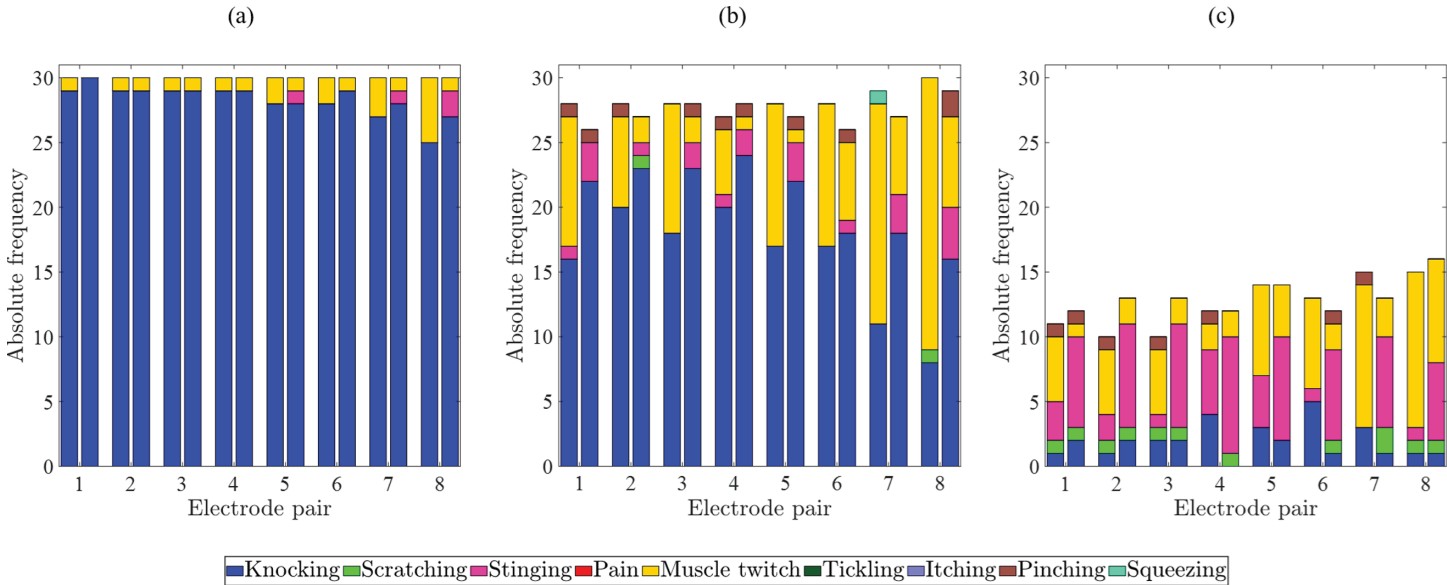

**Fig 4. Qualitative perceptions at perception (a), attention (b), and intolerance thresholds (c) in dependence of electrode pairs and type: TENS (left column) and textile (right column).**

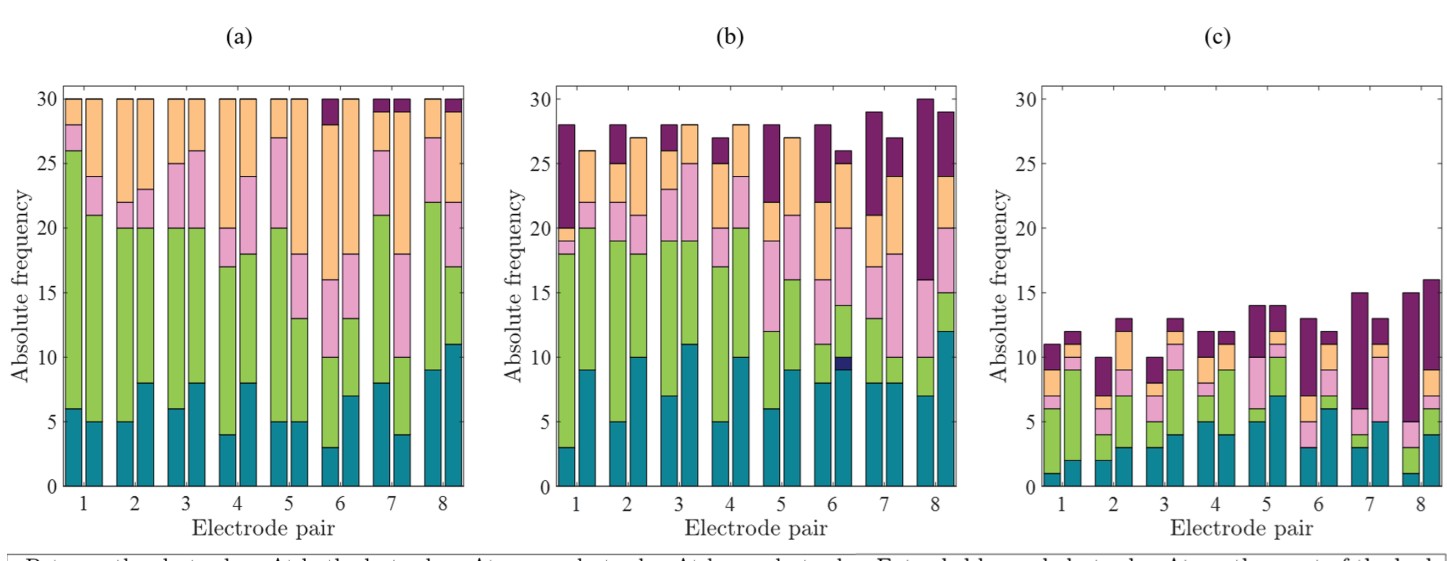

**Fig 5. Spatial perceptions at perception (a), attention (b), and intolerance thresholds (c) in dependence of electrode pairs and type: TENS (left column) and textile (right column).**

Within the study group, the median [25th-75th percentile] pulse interval that led to a vibrating sensation of the electrocutaneous stimulation was 24 [33-19] ms. One participant could not reach $A_v$ (warning amplitude under vibration) as the stimulation was too uncomfortable already at lower amplitudes. The median [25th-75th percentile] values for alertness, discomfort, and urgency were at 7 [6-8], 6 [5-8] and 6 [6-7] at $A_v$. None of the participants had muscle twitches at the amplitude $A_v$. One participant experienced muscle twitches for

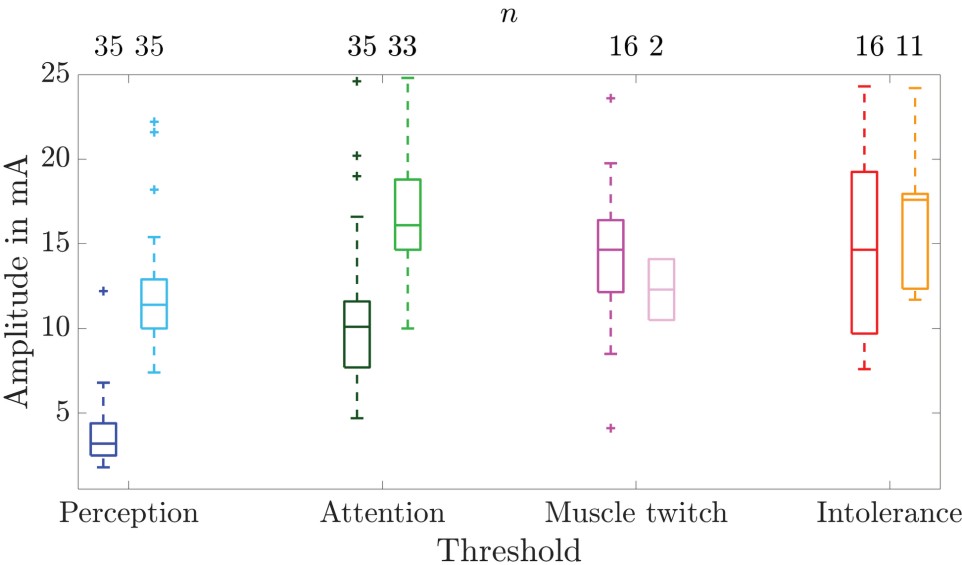

**Fig 6. Boxplots of perception, attention, muscle twitch, and intolerance thresholds at electrode pair 3 of the textile electrodes: without (left box plots & dark colors) and during vibration (right boxplots & bright colors).** The number of participants (*n*) out of 36 is given above each threshold. Note that for the muscle twitch thresholds at the textile cuff the upper and the lower whiskers are equal to the 25th and 75th percentiles, respectively, hence, they are not visible.

amplitudes below $A_v$ which vanished with increasing amplitude. One participant showed muscle twitches for amplitudes larger than $A_v$. The maximal strength of the muscle twitches were 'Visible'. For 21 participants it was possible to increase the stimulation amplitude of the warning pattern up to 25 mA.

The textile cuff was investigated regarding the reliability during the presentation of the warning pattern. For 6 participants, two electrode pairs needed to be rejected from the pattern presentation due to too high impedance at electrode pairs 1, 2, or 7. One electrode pair needed to be rejected for 4 participants including pairs 4, 6, or 7. A pattern presentation with a minimum of 5 neighboring electrode pairs, as determined as lower limit for the presentation of a circumferential warning signal [32], was possible for all cases.

## Discussion

The comparison of TENS and textile electrodes revealed larger thresholds for the TENS electrodes in comparison to the textile electrodes for all electrode pairs at the perception threshold and pairs 5 and 7 at the attention and at pairs 3 and 4 at the intolerance threshold. This result might be explained by the smaller size of the textile electrodes (mean diameter of 20.37 mm) compared to the dimensions of 25 mm×40 mm of the TENS electrodes. According to our previous research, increasing electrode sizes lead to increasing perception thresholds of TENS electrode pair 3 [23], where also attention and intolerance thresholds showed single cases with increased median values for larger electrode sizes compared to smaller ones at electrode pair 3 [23].

The qualitative and spatial perceptions were comparable except for the frequency of muscle twitches and the corresponding report of the spatial perception 'at another part of the body'.

Both studies showed less frequent muscle twitches for the textile electrodes during the threshold determination in direct comparison for the same participants as well as during the presentation of the warning pattern compared to our previous study. The muscles of the upper arm contain primarily fast-twitch muscle fibers (Type II fibers). This type of muscle fibers shows an increase of the twitch tension during hypothermia [36]. Therefore, the application of warm NaCl solution before attaching the textile electrode cuff might have contributed to the decrease of muscle twitch occurrences. We suspect that current distribution within the skin penetrates less deep using the textile electrodes due to the smaller electrode size and the more circular shape leading to a reduced frequency of muscle twitches. Further, the tightening of the cuff might have decreased the frequency of muscle twitching. In our previous study, we compared TENS electrodes vs. three textile electrode pairs placed at the lateral position of the upper right arm [25] where no differences could be found regarding the muscle twitch frequency. In this previous study, it was possible to match the size and shape of TENS and textile electrodes more closely. These assessments are supported by another study [37] in which textile electrodes were used for neuromuscular electrical stimulation of the lower extremities and showed comparable results in terms of comfort, stimulation efficiency, and consistency compared to hydrogel electrodes, while additionally offering advantages such as improved reusability and skin compatibility.

In accordance to our previous work, the perception, attention, and intolerance threshold increased during vibration.

The median reported values of alertness, discomfort, and urgency at warning amplitude $A_v$ determined under vibration are 7, 6, and 6. In our previous study median values were 8, 7, and 8. However, it needs to be noted that these values were evaluated relative to the warning pattern presentation during rest without vibration (all scale rates assigned to 5). In contrast, in study 2, reported here, the values were evaluated relative to the warning pattern presentation at amplitude $A$ under vibration. Nevertheless, the differences might be explained by the fact that 73% of participants that reached the second warning amplitude $A_v$ in our previous study experienced muscle twitches which might lead to higher scaling in comparison to the current study with the textile cuff where no participant showed muscle twitching at $A_v$. Using the textile electrodes, 34 out of 35 participants were able to reach $A_v$ and none of them experienced muscle twitches. The absence of muscle twitch might explain why higher stimulation amplitudes could be tolerated in comparison to our previous study with TENS electrodes. Future studies will focus on the efficacy of the warning in a dual manner to quantify whether a warning signal worked or not.

Despite the application of NaCl solution, there were still cases with too high electrode-skin transition impedance such that a stimulation up to 25 mA was not possible. However, no cases of too high transition impedance occurred for electrode pair no. 3 (S3 Table). A factor that likely influenced this transition impedance is the contact pressure. Due to the manual manufacturing of the textile electrode cuffs, the contact pressure might vary across electrodes. Future studies need to measure the contact pressure around the arm. In a future practical application, the use of NaCl solution is unfeasible. We assume that the remaining impedance problems are rather caused by the small electrode size than the material of the textile electrode. Future work will therefore concentrate on electrode optimization to achieve electrocutaneous warning with low transition impedance and minimal muscle twitching.

Our study has several limitations. The participant group was predominantly young, with an age distribution skewed towards individuals under 40. Future research will aim for a more balanced age range (18-65 years), as age-related effects on perception thresholds have been documented in electrocutaneous stimulation studies [38]. The study sample was primarily university students, which may limit the generalizability of the findings. Future studies will

include a broader participant pool, such as working professionals who are potential users of electrical warning systems.

To assess perception, attention, muscle twitch and intolerance thresholds, a simple method of increasing stimulation amplitude was used instead of more precise methods like the best PEST [39], to maintain a reasonable experiment duration and minimize participant fatigue. The lack of randomization in the presentation order may have introduced sequence effects. Additionally, the absence of blinding in experiment conduction and evaluation may have introduced bias; future work will separate these roles to mitigate this issue.

Additionally, the study focused on specific electrode configurations (vertical pairs) and fixed electrode sizes chosen for practical reasons. Alternative configurations, such as transversal or diagonal pairs, as previous studies showed no benefit regarding muscle twitch frequency or perception of the electrical warning signal [40]. Variability in the impedance of textile electrodes due to manual manufacturing affected comparability between textile electrode pairs of the same cuff and between cuffs of varying sizes, underscoring the need for automated production methods to improve consistency in future studies.

The experimental setup, designed for laboratory-based research, was stationary and unsuitable for field studies or real-world warning scenarios. Future work will focus on developing a portable, potentially wearable version.

## Conclusion

Our study contributes to the ongoing effort to establish an electrocutaneous stimulation system to alert workers in potentially hazardous situations. We successfully demonstrate the application of warning signals through wearable textile electrodes. We compare the novel textile electrode cuff with standard TENS electrodes and demonstrate comparable effectiveness. The textile electrodes showed occasional cases where the electrode-skin transition impedance was too high, which will be addressed through electrode optimization in future work. We found that textile electrodes within the cuff caused less frequent muscle twitches compared to TENS electrodes. To further reduce muscle twitches, future work will optimize the shape and positioning of the electrodes.

We conclude that impedance monitoring is advisable for a future electric warning wearable. In addition, efforts will be made to miniaturize the setup, making it more portable and facilitating the integration of the electrodes into work clothing. Field tests in real work environments will be conducted to assess the system's effectiveness under practical conditions also considering a wider age distribution of participants. Furthermore, long-term applications will be examined, including continuous use over a full 8-hour workday.

Beyond occupational safety, the findings of this and future studies may also be relevant for other applications of electrocutaneous stimulation, such as sensory feedback for prosthetics or muscle stimulation.

## Supporting information

**S1 Fig. Histogram of used cuff sizes within study 1.**
(TIFF)

**S1 Table. Number of participants out of 30 without muscle twitches in dependence of the electrode pair and the electrode type.**
(PDF)

**S2 Table. Number of participants out of 30 with attention and intolerance thresholds >25 mA in dependence of the electrode pair and the electrode type.**
(PDF)

**S3 Table. Number of participants out of 30 and median stimulation amplitudes in mA, where too high transition impedance occurred for textile electrodes in dependence of the threshold and the electrode pair.**
(PDF)

## Author contributions

**Conceptualization:** Eva-Maria Dölker, Tino Kühn, Jens Haueisen.

**Data curation:** Yasemin Cabuk.

**Formal analysis:** Eva-Maria Dölker.

**Investigation:** Eva-Maria Dölker.

**Methodology:** Eva-Maria Dölker, Yasemin Cabuk, Tino Kühn.

**Project administration:** Eva-Maria Dölker, Jens Haueisen.

**Supervision:** Eva-Maria Dölker, Jens Haueisen.

**Validation:** Eva-Maria Dölker.

**Visualization:** Eva-Maria Dölker, Yasemin Cabuk.

**Writing – original draft:** Eva-Maria Dölker.

**Writing – review & editing:** Yasemin Cabuk, Tino Kühn, Jens Haueisen.

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
