## [Decision Letter · Decision Letter 0]

6 Feb 2025

PONE-D-25-02058Comparison of TENS electrodes and textile electrodes for electrocutaneous warningPLOS ONE

Dear Dr. Dölker,

Thank you for submitting your manuscript to PLOS ONE. After careful consideration, we feel that it has merit but does not fully meet PLOS ONE’s publication criteria as it currently stands. Therefore, we invite you to submit a revised version of the manuscript that addresses the points raised during the review process.

We look forward to receiving your revised manuscript.

Kind regards,

Agnese Sbrollini

Academic Editor

PLOS ONE

**Journal Requirements:**

We would like to thank the Free State of Thuringia for its support under project number 2018 IZN 004 co-financed by the European Union under the European Regional Development Fund (ERDF). This work was supported by a scholarship for the professional qualification of young female scientists, funded by the Free State of Thuringia. We acknowledge support for the publication costs by the Open Access Publication Fund of the Technische Universität Ilmenau.  

5. We note that you have referenced "Dolker EM, Grollich D, Schmauder M, Haueisen J." which has currently not yet been accepted for publication. Please remove this from your References and amend this to state in the body of your manuscript: (Dolker EM, Grollich D, Schmauder M, Haueisen J. [Submitted]”) as detailed online in our guide for authors

6. We notice that your supplementary tables are included in the manuscript file. Please remove them and upload them with the file type 'Supporting Information'. Please ensure that each Supporting Information file has a legend listed in the manuscript after the references list.

7. We notice that your supplementary figures are uploaded with the file type 'Figure'. Please amend the file type to 'Supporting Information'. Please ensure that each Supporting Information file has a legend listed in the manuscript after the references list.

Reviewers' comments:

Reviewer's Responses to Questions

**Comments to the Author**

1. Is the manuscript technically sound, and do the data support the conclusions?

Reviewer #1: Yes

Reviewer #2: Yes

2. Has the statistical analysis been performed appropriately and rigorously? 

Reviewer #1: Yes

Reviewer #2: Yes

3. Have the authors made all data underlying the findings in their manuscript fully available?

Reviewer #1: Yes

Reviewer #2: Yes

4. Is the manuscript presented in an intelligible fashion and written in standard English?

Reviewer #1: Yes

Reviewer #2: Yes

5. Review Comments to the Author

**Reviewer #1:** 1.Kindly add the future scope of the work

2. To enhance the document, it is recommended to incorporate a detailed section on the future scope of the work. This addition should explore potential advancements, applications, or extensions of the study, emphasizing how it could address existing gaps or open new research avenues. For instance, the future scope could highlight innovative methodologies, interdisciplinary collaborations, or emerging technologies that could further develop the study's findings.

3.Additionally, it is crucial to update the reference section to include research papers cited from the year 2025. These references should predominantly draw from studies conducted during 2023–2024 to ensure the document reflects the latest advancements and aligns with contemporary scholarly discourse. Integrating these citations will substantiate the relevance of the work and demonstrate its grounding in the most recent research trends.

**Reviewer #2: **1- The research gap in the abstract was not well defined, in addition to what are the standards and what are their values that the researcher obtained during the work.

2- Identify the researcher’s contributions in brief and clear points, in addition to including a paragraph at the end of the introduction that explains the structure of the research in all its sections.

3- Clarify future work in the conclusions, in addition to relying on numerical values for the results in building clear conclusions for the reader.

4- Arrange references in one format

6. PLOS authors have the option to publish the peer review history of their article (what does this mean?). If published, this will include your full peer review and any attached files.

Reviewer #1: No

Reviewer #2: No

---

## [Author Response · Author response to Decision Letter 1]

24 Feb 2025

Dear Dr. Agnese Sbrollini and Reviewers,

On behalf of my co-authors and myself, I extend our sincere thanks for your time and effort in reviewing our manuscript titled “Comparison of TENS electrodes and textile electrodes for electrocutaneous warning”. Your detailed comments and constructive feedback have been very helpful in improving our work. We appreciate your thorough evaluations and professional guidance.

Please find our responses in the attached rebuttal letter.

Best regards,

Eva-Maria Dölker

---

## [Decision Letter · Decision Letter 1]

18 Mar 2025

PONE-D-25-02058R1Comparison of TENS electrodes and textile electrodes for electrocutaneous warningPLOS ONE

Dear Dr. Dölker,

Thank you for submitting your manuscript to PLOS ONE. After careful consideration, we feel that it has merit but does not fully meet PLOS ONE’s publication criteria as it currently stands. Therefore, we invite you to submit a revised version of the manuscript that addresses the points raised during the review process.

We look forward to receiving your revised manuscript.

Kind regards,

Agnese Sbrollini

Academic Editor

PLOS ONE

Journal Requirements:

Reviewers' comments:

Reviewer's Responses to Questions

**Comments to the Author**

1. If the authors have adequately addressed your comments raised in a previous round of review and you feel that this manuscript is now acceptable for publication, you may indicate that here to bypass the “Comments to the Author” section, enter your conflict of interest statement in the “Confidential to Editor” section, and submit your "Accept" recommendation.

Reviewer #1: All comments have been addressed

Reviewer #2: All comments have been addressed

2. Is the manuscript technically sound, and do the data support the conclusions?

Reviewer #1: Yes

Reviewer #2: (No Response)

3. Has the statistical analysis been performed appropriately and rigorously? 

Reviewer #1: Yes

Reviewer #2: (No Response)

4. Have the authors made all data underlying the findings in their manuscript fully available?

Reviewer #1: Yes

Reviewer #2: (No Response)

5. Is the manuscript presented in an intelligible fashion and written in standard English?

Reviewer #1: Yes

Reviewer #2: (No Response)

6. Review Comments to the Author

Reviewer #1: (No Response)

Reviewer #2: (No Response)

7. PLOS authors have the option to publish the peer review history of their article (what does this mean?). If published, this will include your full peer review and any attached files.

Reviewer #1: **Yes: **Dr.Alankrita Aggarwal

Reviewer #2: No

---

## [Author Response · Author response to Decision Letter 2]

24 Mar 2025

Dear Dr. Agnese Sbrollini and Reviewers,

On behalf of my co-authors and myself, I extend our sincere thanks for your time and effort in reviewing the revised manuscript titled “Comparison of TENS electrodes and textile electrodes for electrocutaneous warning”. We appreciate your thorough evaluations and professional guidance.

Please find our responses in the attached "Response to the Reviewers" document.

Best regards,

Dr. Eva-Maria Dölker

---

## [Decision Letter · Decision Letter 2]

9 Apr 2025

Comparison of TENS electrodes and textile electrodes for electrocutaneous warning

PONE-D-25-02058R2

Dear Dr. Dölker,

We’re pleased to inform you that your manuscript has been judged scientifically suitable for publication and will be formally accepted for publication once it meets all outstanding technical requirements.

Kind regards,

Agnese Sbrollini

Academic Editor

PLOS ONE

Additional Editor Comments (optional):

Reviewers' comments:

Reviewer's Responses to Questions

**Comments to the Author**

1. If the authors have adequately addressed your comments raised in a previous round of review and you feel that this manuscript is now acceptable for publication, you may indicate that here to bypass the “Comments to the Author” section, enter your conflict of interest statement in the “Confidential to Editor” section, and submit your "Accept" recommendation.

Reviewer #1: All comments have been addressed

Reviewer #2: (No Response)

2. Is the manuscript technically sound, and do the data support the conclusions?

Reviewer #1: Yes

Reviewer #2: (No Response)

3. Has the statistical analysis been performed appropriately and rigorously? 

Reviewer #1: Yes

Reviewer #2: (No Response)

4. Have the authors made all data underlying the findings in their manuscript fully available?

Reviewer #1: Yes

Reviewer #2: (No Response)

5. Is the manuscript presented in an intelligible fashion and written in standard English?

Reviewer #1: Yes

Reviewer #2: (No Response)

6. Review Comments to the Author

Reviewer #1: User Experience Evaluation: More detailed subjective feedback i.e from surveys or interviews could strengthen the case for textile electrode adoption if user comfort is addressed,

Long-Term Use Implications: The paper briefly touches on durability and wearability, but further exploration of long-term usability or wear resistance would be valuable.

Environmental Factors to test the performance of textile electrodes in varying humidity or sweat conditions is not extensively tested, though it may affect signal reliability.

Reviewer #2: (No Response)

7. PLOS authors have the option to publish the peer review history of their article (what does this mean?). If published, this will include your full peer review and any attached files.

Reviewer #1: **Yes: **Dr. Alankrita Aggarwal

Reviewer #2: No

---

## [Editor Report · Acceptance letter]

PONE-D-25-02058R2

PLOS ONE

Dear Dr. Dölker,

I'm pleased to inform you that your manuscript has been deemed suitable for publication in PLOS ONE. Congratulations! Your manuscript is now being handed over to our production team.

Kind regards,

on behalf of

Dr. Agnese Sbrollini

Academic Editor

PLOS ONE